# ReLLa: Retrieval-enhanced Large Language Models for Lifelong Sequential Behavior Comprehension in Recommendation

## ABSTRACT

With large language models (LLMs) achieving remarkable break-throughs in natural language processing (NLP) domains, LLM-enhanced recommender systems have received much attention and have been actively explored currently. In this paper, we focus on adapting and empowering a pure large language model for zero-shot and few-shot recommendation tasks. First and foremost, we identify and formulate the *lifelong sequential behavior incomprehension problem* for LLMs in recommendation domains, *i.e.*, LLMs fail to extract useful information from a textual context of long user behavior sequence, even if the length of context is far from reaching the context limitation of LLMs. To address such an issue and improve the recommendation performance of LLMs, we propose a novel framework, namely **R**etrieval-**e**nhanced **L**arge **La**nguage models (ReLLa) for recommendation tasks in both zero-shot and few-shot settings. For zero-shot recommendation, we perform semantic user behavior retrieval (SUBR) to improve the data quality of testing samples, which greatly reduces the difficulty for LLMs to extract the essential knowledge from user behavior sequences. As for few-shot recommendation, we further design retrieval-enhanced instruction tuning (ReiT) by adopting SUBR as a data augmentation technique for training samples. Specifically, we develop a mixed training dataset consisting of both the original data samples and their retrieval-enhanced counterparts. We conduct extensive experiments on three real-world public datasets to demonstrate the superiority of ReLLa compared with existing baseline models, as well as its capability for lifelong sequential behavior comprehension. **To be highlighted, with only less than 10% training samples, *few-shot* ReLLa can outperform traditional CTR models that are trained on the entire training set (*e.g.,* DCNv2, DIN, SIM).** The code is available for the reviewers[1].

## 1 INTRODUCTION

Recommender systems play a vital role in various online applications to alleviate the information overload problem and satisfy the users' information needs [23, 78, 79]. Besides, large language models (LLMs) have flourished in the natural language processing (NLP) domain, showing impressive capacities in generating human-like texts for a wide range of tasks [4, 73]. Consequently, recent works start to explore the potential of LLMs for recommender systems [1, 29, 44]. They adopt LLMs directly for various recommendation tasks (*e.g.*, listwise ranking, pointwise scoring), and find out that large language models depict promising performance in zero-shot and few-shot settings for recommendation [1, 86].

In this paper, we focus on adapting and empowering a pure large language model for recommendation tasks in zero-shot and few-shot settings. First, we identify the **lifelong sequential behavior incomprehension problem**, *i.e.*, *LLMs fail to extract the useful information from a textual context of long user behavior sequence*

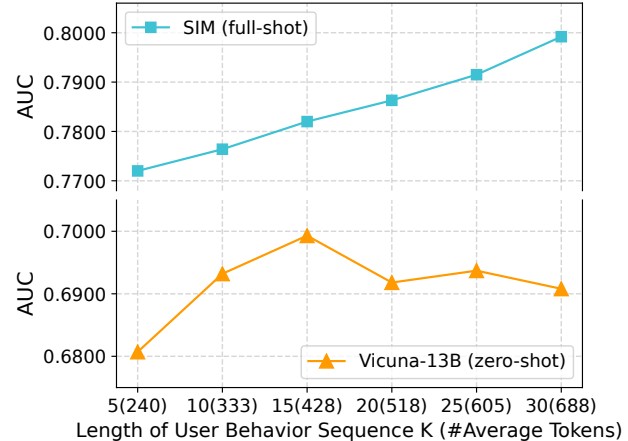

**Figure 1: The illustration of lifelong sequential behavior incomprehension problem for LLMs. We report the AUC performance of SIM and Vicuna-13B on MovieLens-1M dataset. While SIM enjoys steady performance improvement as the length of behavior sequence $K$ grows, Vicuna-13B only peaks at $K = 15$ and fails to extract the useful information with further longer sequences (*i.e.,* $K > 15$).**

*for recommendation tasks, even if the length of context is far from reaching the context limitation of LLMs.* This problem is shown in Figure 1, where Vicuna-13B [10, 73] is a popular open-source large language model with a context window of 2048 tokens. As we can observe, the traditional recommendation model (*i.e.*, SIM) enjoys steady performance gains as the length of involved user sequence $K$ grows. However, the performance of Vicuna-13B reaches the peak at length $K = 15$ and starts to decrease with longer behavior sequence $K > 15$, even if the number of involved tokens is far less than the context window limitation (*i.e.*, 2048 tokens). While in common NLP tasks, LLMs can definitely exhibit exceptional performance if given a similar length of context (around 600+ tokens). Therefore, we argue that such an incomprehension problem on long user behavior sequence is special for LLMs in recommendation domains, where it is a rather difficult reasoning task to infer the user's preference towards a certain candidate item based on the given user profile and behavior history.

To address the lifelong sequential behavior incomprehension problem, we propose a novel framework to develop **R**etrieval-**e**nhanced **L**arge **La**nguage models (ReLLa) for recommendation tasks in both zero-shot and few-shot settings. For *zero-shot recommendation*, we propose to conduct semantic user behavior retrieval (SUBR) to replace the simply truncated top-$K$ recent behaviors with the top-$K$ semantically relevant behaviors towards the target item. In this way, we improve the quality of data samples and reduce the difficulty for LLMs to extract useful information from user behavior sequences, therefore alleviating the incomprehension problem. For *few-shot recommendation*, apart from applying SUBR to improve the

---

[1]https://anonymous.4open.science/r/ReLLa/

data quality of samples, we propose to perform retrieval-enhanced instruction tuning (ReiT) to further promote the ability of LLMs to handle inputs with long behavior sequences. We apply SUBR on training samples as the data augmentation techniques to obtain a mixed training dataset of both original and retrieval-enhanced training data samples, which increases the robustness and generalization ability of LLMs. **More surprisingly, with only few-shot training samples (*e.g.*, 8,192 data instances in MovieLens-25M dataset), ReLLa can outperform full-shot traditional recommendation models (*e.g.*, DCNv2 [77], DIN [92], and SIM [59]) that are trained with the entire training set (*e.g.*, nearly 20M samples in MovieLens-25M dataset).**

Main contributions of this paper are in three folds:

- To the best of our knowledge, we are the first to identify and well formulate the lifelong sequential behavior incomprehension problem for LLMs in recommendation, where LLMs are generally incomprehensible to a textual context of long user behavior sequence, even if the length of context is far from reaching the context limitation.
- We propose a novel ReLLa (**R**etrieval-**e**nhanced **L**arge **La**nguage Models) framework to mitigate the incomprehension problem of LLMs on long user behavior sequences. We design semantic user behavior retrieval (SUBR) to improve the data quality of data samples for zero-shot recommendation, and further propose retrieval-enhanced instruction tuning (ReiT) to promote the few-shot recommendation performance with a mixture of original and retrieval-enhanced training samples.
- Extensive experiments on three real-world public datasets validate the effectiveness of our method compared with existing baselines. **Note that the baseline models are trained in *full-shot* settings with the entire training set, while ReLLa is only trained with *few-shot* samples.**

## 2 PRELIMINARIES

In this paper, we focus on the click-through rate (CTR) prediction, which serves as the core component in recommender systems to estimate a user's click probability towards a target item given a certain context [23, 46]. The training dataset for CTR prediction is denoted as $\{(x_i, y_i)\}_{i=1}^N$, where $N$ is the number of data samples (*i.e.*, $N$-shot). When adapting a pure large language model for such a pointwise scoring task, we need to clarify the following three key aspects: (1) what is the definition of zero-shot and few-shot recommendations, (2) how to formulate the textual input-output pairs, and (3) how to do pointwise scoring with LLMs.

### 2.1 Zero-shot and Few-shot Recommendations

Zero-shot recommendation implies that a model is directly employed for the target recommendation task without any tuning on the in-domain training data. Apparently, traditional recommendation models are incapable of accomplishing zero-shot recommendation tasks, since they are randomly initialized. However, LLMs possess a vast volume of open-world knowledge and logical reasoning abilities, which enable them to infer the user's preference towards a certain target item based on the profile of user/item.

Few-shot recommendation refers to low-resource scenarios with $N$ training data samples. $N$ denotes the number of shots, which is

a relatively small number. This highly requires the data efficiency characteristic of an algorithm to fully exploit the limited number of training samples to achieve better recommendation performance.

Extending from the definition of few-shot recommendation, we can therefore define full-shot recommendation as the setting where we train the model based on the entire training set.

### 2.2 Textual Input-Output Pair Formulation

For LLMs, we need to convert each data sample $x_i$ into textual sentences $x_i^{text}$ via hard prompt templates. Similarly, the binary label $y_i \in \{0, 1\}$ is transformed into a pair of binary key answer words $y_i^{text} \in \{\text{"Yes", "No"}\}$. We give an illustrative example of the input-output pair $(x_i^{text}, y_i^{text})$ in Figure 2, where $x_i^{text}$ contains the descriptive texts for user profile, user behavior sequence, target item and task description, respectively.

> **Input:**
> The user is a male. His job is college/grad student. His age is 25-34.
> He watched the following movies in order in the past, and rated them:
> ['0. Pump Up the Volume (1990) (4 stars)', '1. Antz (1998) (4 stars)', "2. Devil's Own, The (1997) (5 stars)", '3. Crying Game, The (1992) (1 star)']
> Based on the movies he has watched, deduce if he will like the movie ***Titanic (1997)***.
> Note that more stars the user rated the movie, the user like the movie more.
> You should ONLY tell me yes or no.
> **Output:**
> No.

**Figure 2: Illustration of textual input-output pair.**

Notably, the predominant factor that determines the length of context is derived from the user behavior sequence, the length of which can varies from tens to hundreds. For each input $x_i$, we truncate the user behavior sequence to length $K$. For example, the length of behavior sequence in Figure 2 is $K = 4$. While the common sequential CTR prediction settings usually truncate and adopt *the most recent $K$ behaviors*, ReLLa propose to conduct semantic user behavior retrieval to construct textual inputs with *the most relevant $K$ behaviors* towards the target item.

### 2.3 Pointwise Scoring with LLMs

The large language model takes as input the discrete tokens of $x_i^{text}$, and generate the next token $\hat{y}_i^{text}$ as the output, the process of which can be formulated as follows:

$$s_i = \text{LLM}(x_i^{text}) \in \mathbb{R}^V,$$
$$p_i = \text{Softmax}(s_i) \in \mathbb{R}^V, \quad (1)$$
$$\hat{y}_i^{text} \sim p_i,$$

where $V$ is the vocabulary size, and $\hat{y}_i^{text}$ is the next predicted token sampled from the probability distribution $p_i$.

However, CTR prediction requires the model to do pointwise scoring, and the output should be floating-point number $\hat{y}_i \in [0, 1]$ instead of a discrete token $\hat{y}_i^{text}$. Therefore, following previous works [1, 90], we intercept the estimated scores $s_i \in \mathbb{R}^V$, and conduct a bidimensional softmax over the corresponding scores of the binary key answer words. Suppose the vocabulary indices for "Yes" and "No" are $a$ and $b$, respectively. The pointwise scoring of LLMs for CTR prediction can be written as:

$$\hat{y}_i = \frac{\exp(s_{i,a})}{\exp(s_{i,a}) + \exp(s_{i,b})} \in (0, 1). \quad (2)$$

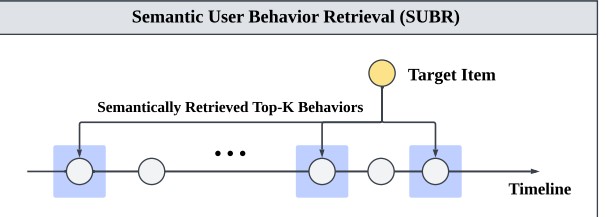

**Figure 3: Illustration of semantic user behavior retrieval (SUBR), which improves the data quality by retrieving the top-$K$ semantically relevant behaviors towards the target item. This reduces the difficulty for LLMs to extract useful information from the user history, and therefore alleviates the long user behavior sequence incomprehension problem.**

It is worth noting that such an estimated click-through rate $\hat{y}_i$ is only leveraged for evaluation on the testing set. We preserve the common instruction tuning and causal language modeling paradigm for LLMs if training is involved.

## 3 METHODOLOGY

In this section, we introduce the ReLLa (**R**etrieval-**e**nhanced **L**arge **La**nguage Models) framework in details.

### 3.1 Overview of ReLLa

In the ReLLa framework, we develop two key techniques for LLMs in zero-shot and few-shot recommendations, respectively.

For zero-shot recommendation, as illustrated in Figure 3, we propose to conduct semantic user behavior retrieval (SUBR) to improve the data quality of data samples. We first leverage the large language model to obtain the semantic vectors for each item. Then, for each textual data sample $x_i^{text}$, we retrieve the most *semantically relevant* $K$ behaviors, which can substitute the original most *recent* $K$ behaviors.

For few-shot recommendation, as shown in Figure 5, we propose to perform retrieval-enhanced instruction tuning (ReiT) to promote the ability of LLMs to extract useful information from long behavior sequences. Notably, the semantic user behavior retrieval (SUBR) is adopted as the data augmentation technique to form the mixed training dataset. The mixture of both original and retrieval-enhanced data samples introduces more variety and patterns in the training set, thus increasing the robustness and generalization ability of LLMs for lifelong sequential behavior comprehension.

Although ReLLa is tuned in ***few-shot*** settings, we would like to again emphasize that other recommendation baseline models are trained in ***full-shot*** settings with the entire training set.

### 3.2 Semantic User Behavior Retrieval

In zero-shot settings, the parameters of LLMs cannot be tuned according to the in-domain training samples. Hence, as shown in Figure 3, semantic user behavior retrieval (SUBR) aims to improve the quality of each sample by replacing the simply truncated most recent $K$ behaviors with the most semantically relevant $K$ behaviors towards the target item. As suggested in previous works [59, 61], the retrieved user behaviors can denoise the user history and convey more clear and essential user interests for the target item, while preserving the original length of user sequence as the model input.

> Here is a movie. Its title is Toy Story (1995). The movie's genre is Animation.

**Figure 4: Illustration of descriptive text for an item (movie).**

Firstly, we conduct semantic item encoding to obtain the semantic vector for each item. For the $t$-th item in the pool, a descriptive text is constructed via hard prompt template (an example is given in Figure 4, and is then fed through LLM. We perform average pooling over all the hidden states from the last layer of LLM, resulting in a vector $u_t \in \mathbb{R}^D$, where $D$ is the hidden size of LLM (*e.g.*, 4096 for Vicuna-7B, and 5120 for Vicuna-13B). A principal component analysis (PCA) [69] module is further employed for both dimension reduction and denoising purposes, engendering the final semantic representation $v_t \in \mathbb{R}^d$, where we set $d = 512$. Now we can measure the semantic relevance between each pair of items via the cosine similarity between their corresponding semantic representations.

Next, we can apply semantic user behavior retrieval on each testing sample to replace the original truncated top-$K$ recent behaviors with the top-$K$ semantically relevant behaviors towards the target item. In this way, we obtain a parallel retrieval-enhanced testing dataset with higher data quality, while keeping the length of input context roughly unchanged. Therefore, SUBR can improve the zero-shot recommendation performance, and mitigate the incomprehension problem on long user behavior sequences.

### 3.3 Retrieval-enhanced Instruction Tuning

As for few-shot recommendation, we denote the training dataset as $\{(x_i^{text}, y_i^{text})\}_{i=1}^N$, where $N$ is the number of shots (*i.e.*, training samples). While previous works [1, 86] directly employ instruction tuning for LLMs over the converted textual input-output pairs, we argue that simple instruction tuning could potentially expose large language models to risks of overfitting and catastrophic forgetting on limited number of training data [35, 66].

To this end, we propose a novel retrieval-enhanced instruction tuning (ReiT), where semantic user behavior retrieval (SUBR) is adopted as the data augmentation technique to construct a mixed training dataset with enriched user behavior patterns. As shown in Figure 5, we apply SUBR on each training data to obtain its retrieval-enhanced counterpart $\tilde{x}_i^{text}$. Next, we merge the original and retrieval-enhanced data instances to construct a mixed training dataset with total $2N$ samples. Finally, we conduct instruction tuning for LLMs on the mixed training data. The pattern enrichment brought by SUBR can regularize and prevent the large language model from overfitting, thus promoting its robustness and generalization ability to effectively extract essential knowledge from a long user behavior sequence of length $K$.

We leverage the causal language modeling objective for instruction tuning to retain the original model structure:

$$\max_{\Theta} \sum_{(x,y)\in\mathcal{M}} \sum_{j=1}^{|y|} \log P_{\Theta}(y_j|x, y_{<j}), \qquad (3)$$

**Figure 5: Illustration of retrieval-enhanced instruction tuning, where we construct a mixed training dataset. The mixed dataset consists of both the original textual input-output samples and their retrieval-enhanced counterparts obtained via semantic user behavior retrieval (SUBR).**

where $\Theta$ is the parameter of LLM, $\mathcal{M}$ is the mixed training dataset with total $2N$ data samples, $y_j$ is the $j$-th token of the textual output $y$, and $y_{<j}$ denotes the tokens before $y_j$. There is *no* randomly initialized prediction layer appended upon LLM for CTR prediction with binary cross-entropy (BCE) loss. The CTR estimation method for pointwise scoring with LLMs discussed in Section 2.3 is only used for evaluation on the testing set.

While we maintain a mixed training dataset for instruction tuning, the testing set contains pure retrieval-enhanced data samples generated by SUBR, which is the same as zero-shot recommendation as described in Section 3.2. Moreover, we provide further discussion about ReiT to address readers' possible concerns as follows:

- *Will ReiT cause the inconsistency between the training and testing data?* Data augmentation is a common regularization technique, especially for low-resource few-shot settings in computer vision (CV) [2, 84] or natural language processing (NLP) [17, 36]. The inconsistency would not exist, as long as the augmentation algorithm is sound and reasonable.
- *Which factor actually contribute to the performance improvement of ReiT? The doubled training samples, or the pattern enrichment?* Both factors can lead to the final performance enhancement, but we argue that the pattern enrichment as regularization is a more important factor for model robustness. Empirical studies are provided in Section 4.5 to ablate and decouple these two factors.

## 4 EXPERIMENT

In this section, we conduct extensive experiments to answer the following research questions:

**RQ1** How does ReLLa perform compared to existing baselines?
**RQ2** Does ReLLa promote the lifelong sequential behavior comprehension ability of LLMs for recommendation tasks?
**RQ3** How does the number of shots $N$ affect the performance?
**RQ4** What are the influences of different components for ReLLa?
**RQ5** How ReLLa help LLMs to better comprehend the user behavior sequence?

## 4.1 Experiment Setup

*4.1.1 Datasets.* We conduct experiments on three real-world datasets (*i.e.*, BookCrossing[2], MovieLens-1M[3] and MovieLens-25M[4]). We

---

[2]http://www2.informatik.uni-freiburg.de/~cziegler/BX/
[3]https://grouplens.org/datasets/movielens/1m/
[4]https://grouplens.org/datasets/movielens/25m/

**Table 1: The dataset statistics.**

| Dataset | #Users | #Items | #Samples | #Fields | #Features |
|---------|--------|--------|----------|---------|-----------|
| BookCrossing | 278,858 | 271,375 | 17,714 | 10 | 912,279 |
| MovieLens-1M | 6,040 | 3,706 | 970,009 | 10 | 16,944 |
| MovieLens-25M | 162,541 | 59,047 | 25,000,095 | 6 | 280,576 |

show the dataset statistics in Table 1 and give detailed data preprocessing information in Appendix A due to page limitations.

*4.1.2 Evaluation Metrics.* To evaluate the performance of our methods, we leverage AUC (area under the ROC curve), Log Loss (binary cross-entropy loss) and ACC (accuracy score) as the evaluation metrics. In CTR prediction, slightly higher AUC or lower Log Loss (e.g., 0.001) can be regarded as significant improvement [43, 77].

*4.1.3 Baseline Models.* The CTR baseline models can be mainly classified into two categories: (1) *traditional CTR models* that take one-hot encoded IDs as inputs, and (2) *LM-based models* that incorporate pretrained language models and formulate CTR prediction as either text classification or sequence-to-sequence problem.

Traditional CTR models can be further categorized into (1) feature interaction models, and (2) user behavior models. We select DeepFM [23], AutoInt [70], and DCNv2 [77] as representative feature interaction models, and choose GRU4Rec [26], Caser [72], SASRec [33], DIN [92], and SIM [59] as representative user behavior models. We apply average pooling over users' historical behaviors, and regard the outputs as additional feature fields for the feature interaction models. SIM [59] is a classical sequential CTR model that leverages user behavior retrieval techniques to enhance the recommendation performance. We include it for fair comparison, since ReLLa incorporates semantic user behavior retrieval (SUBR). As for LM-based CTR models, we select CTR-BERT [53], PTab [48], and P5 [20] as the representative baselines. TALLRec [1] adopts the simple instruction tuning framework for LLMs, and we therefore include it in our ablation study in Section 4.5.

*4.1.4 Implementation Details.* We select Vicuna-13B [10] released by FastChat[5] as the base LLM for ReLLa. All the experiments are conducted on V100 GPUs. For training resource efficiency, 8-bit quantization and low-rank adaption (LoRA) [30] are adopted for parameter-efficient finetuning (PEFT). We follow previous works [1, 9] to set the configuration of LoRA, with LoRA rank as 8, LoRA alpha as 16, and LoRA dropout as 0.05. The LoRA update matrices are applied on the query and value projection matrices of attention blocks. During instruction tuning, we adopt AdamW [51] optimizer with weight decay set to 0. The model is trained with a batch size selected from $\{128, 256\}$. The learning rate is initialized from $\{1 \times 10^{-3}, 1.5 \times 10^{-3}\}$ with linear scheduler. On BookCrossing dataset, the maximum training epoch is set to 10, while on MovieLens-1M and MovieLens-25M datasets, the maximum epoch is set to 5. The configuration of baselines is in Appendix B. The hard prompt templates for textual input-output pairs and item descriptions for all three datasets are in Appendix C.

Moreover, when constructing the hard prompt template for ReLLa, we remove all the pure ID fields, *i.e.*, *User ID* and *ISBN* fields on BookCrossing dataset, *User ID*, *Movie ID*, and *Zipcode* fields on MovieLens-1M dataset, *User ID* and *Movie ID* fields on MovieLens-25M dataset. The reason is that LLMs possess limited

---

[5]https://github.com/lm-sys/FastChat

**Table 2: The performance of different models in *zero-shot*, *full-shot* and *few-shot* settings. In *full-shot* setting, the baselines are trained on the entire training set. In *few-shot* setting, the number of training shots $N$ is selected from $\{256(<1\%), 1024(<10\%)\}$ on BookCrossing dataset, and $\{8192(<1\%), 65536(<10\%)\}$ on MovieLens-1M and MovieLens-25M datasets. The best result is given in bold, and the second-best value is underlined. *Rel.Impr* denotes the relative AUC improvement rate of ReLLa against each baseline. The symbol $*$ indicates statistically significant improvement of ReLLa over the best baseline with $p$-value < 0.001.**

| Model | | BookCrossing | | | | MovieLens-1M | | | | MovieLens-25M | | | |
|---|---|---|---|---|---|---|---|---|---|---|---|---|---|
| | | AUC | Log Loss | ACC | Rel.Impr | AUC | Log Loss | ACC | Rel.Impr | AUC | Log Loss | ACC | Rel.Impr |
| Zero-shot | Vicuna-7B | 0.7011 | 0.9357 | 0.5378 | 3.45% | 0.6739 | 0.9510 | 0.5644 | 4.07% | 0.7468 | 0.6348 | 0.6392 | -1.93% |
| | Vicuna-13B | 0.7176 | 0.9507 | 0.5649 | 1.07% | 0.6993 | 0.6291 | 0.6493 | 0.29% | 0.7503 | 0.6308 | 0.6427 | -2.39% |
| | ReLLa (Ours) | 0.7253* | 0.9277* | 0.5750* | - | 0.7013* | 0.6250* | 0.6507* | - | 0.7324 | 0.5858* | 0.7027* | - |
| Full-shot | DeepFM | 0.7496 | 0.5953 | 0.6760 | 1.05% | 0.7915 | 0.5484 | 0.7225 | 1.49% | 0.8189 | 0.4867 | 0.7709 | 3.52% |
| | AutoInt | 0.7481 | 0.6840 | 0.6365 | 1.26% | 0.7929 | 0.5453 | 0.7226 | 1.31% | 0.8169 | 0.4957 | 0.7689 | 3.77% |
| | DCNv2 | 0.7472 | 0.6816 | 0.6472 | 1.38% | 0.7931 | 0.5464 | 0.7216 | 1.29% | 0.8190 | 0.4989 | 0.7702 | 3.50% |
| | GRU4Rec | 0.7479 | 0.5930 | 0.6777 | 1.28% | 0.7926 | 0.5453 | 0.7225 | 1.35% | 0.8186 | 0.4941 | 0.7700 | 3.55% |
| | Caser | 0.7478 | 0.5990 | 0.6760 | 1.30% | 0.7918 | 0.5464 | 0.7206 | 1.45% | 0.8199 | 0.4865 | 0.7707 | 3.39% |
| | SASRec | 0.7482 | 0.5934 | **0.6811** | 1.24% | 0.7934 | 0.5460 | 0.7233 | 1.25% | 0.8187 | 0.4956 | 0.7691 | 3.54% |
| | DIN | 0.7477 | 0.6811 | 0.6557 | 1.31% | 0.7962 | 0.5425 | 0.7252 | 0.89% | 0.8190 | 0.4906 | 0.7716 | 3.50% |
| | SIM | 0.7541 | **0.5893** | 0.6777 | 0.45% | 0.7992 | 0.5387 | 0.7268 | 0.51% | 0.8344 | 0.4724 | 0.7822 | 1.59% |
| | CTR-BERT | 0.7448 | 0.5938 | 0.6704 | 1.71% | 0.7931 | 0.5457 | 0.7233 | 1.29% | 0.8079 | 0.5044 | 0.7511 | 4.93% |
| | PTab | 0.7429 | 0.6154 | 0.6574 | 1.97% | 0.7955 | 0.5428 | 0.7240 | 0.98% | 0.8107 | 0.5022 | 0.7551 | 4.56% |
| | P5 | 0.7438 | 0.6128 | 0.6563 | 1.84% | 0.7937 | 0.5478 | 0.7190 | 1.21% | 0.8092 | 0.5030 | 0.7527 | 4.76% |
| Few-shot | ReLLa (<1%) | 0.7482 | 0.6265 | 0.6800 | - | 0.7927 | 0.5475 | 0.7196 | - | 0.8352 | 0.4693 | 0.7779 | - |
| | ReLLa (<10%) | **0.7575*** | 0.5919 | 0.6806 | - | **0.8033*** | **0.5362*** | **0.7280*** | - | **0.8477*** | **0.4524*** | **0.7925*** | - |

perceptual abilities for pure ID texts [44]. Other fields are leveraged as user profile or item information in the prompt, as described in Section 2.2 and Appendix C. Note that we ***do not*** discard any features for other CTR baseline models, *i.e.*, they take all the feature fields and user behavior sequences as inputs.

## 4.2 Overall Performance (RQ1)

We evaluate the performance of ReLLa in comparison to existing baseline models, and report the results in Table 2. Note that other recommendation baseline models are all trained in ***full-shot*** settings with the entire training set. We set the length of user behavior sequence $K$ to 60/30/30 for BookCrossing/MovieLens-1M/MovieLens-25M, respectively.

For zero-shot recommendation, we observe that:

- The performance of Vicuna-7B is notably inferior to its 13B version on all three datasets. It demonstrates that a larger LLM possesses more excellent language comprehension and logical reasoning abilities, therefore leading to better zero-shot inference capability for user preference.
- ReLLa significantly outperforms Vicuna-13B for all three metrics on BookCrossing and MovieLens-1M datasets. Although the AUC performance of ReLLa degenerates on MovieLens-25M, ReLLa attains significant improvements in terms of pointwise metrics (*i.e.*, Log Loss and ACC). such phenomena validate the effectiveness of SUBR in reducing the difficulty for LLMs to extract useful information from user behavior sequences. Also, the AUC degeneration of AUC on MovieLens-25M reveals the potential instability of zero-shot LLMs for recommendation.

As for full-shot and few-shot settings, we can draw the following observations from Table 2:

- SIM achieves the best performance among all the baseline models. SIM applies user behavior retrieval to reduce the noise of

user sequences, which is essentially beneficial for CTR prediction. Besides, LM-based CTR models (*i.e.*, CTR-BERT, PTab, P5) perform worse than most of the ID-based traditional CTR models, which is consistent with the results reported in [40, 65]. These LM-based methods only incorporate small language models (*e.g.*, BERT [2], T5 [64]) for pure text-based recommendation, and therefore result in inferior performance.
- ReLLa (few-shot) generally achieves significantly better performance over all the baseline models, except for few cases, which validates the effectiveness of our proposed retrieval-enhanced instruction tuning (ReiT). **It is worth noting that ReLLa only utilizes less than 10% training samples for finetuning, while other baseline models are trained on the entire training set**, *e.g.*, $N = 65,536$ for ReLLa and $N = 19,349,912$ for SIM on MovieLens-25M dataset. This demonstrates the superior data efficiency of ReLLa for sequential recommendation tasks.

## 4.3 Sequential Behavior Comprehension (RQ2)

We vary the length of user behavior sequence $K$ to investigate its impact on CTR prediction performance, which can demonstrate the comprehension ability of a model towards user behavior sequences. Three different models, including SIM (full-shot), Vicuna-13B (zero-shot) and ReLLa (few-shot), are evaluated with different $K$s. On BookCrossing dataset, $K$ ranges in $\{10, 20, 30, 40, 50, 60\}$, while on MovieLens-1M and MovieLens-25M datasets, $K$ ranges in $\{5, 10, 15, 20, 25, 30\}$. The numbers of shots are set to 256, 8192, 8192 for BookCrossing, MovieLens-1M, and MovieLens-25M, respectively (*i.e.*, <1% few-shot setting). The results are shown in Figure 6, from which we obtain the following observations:

- As a traditional CTR prediction model, SIM (full-shot) [59] enjoys steady performance improvement as the length $K$ grows. This is consistent with our common understanding, where longer user behavior sequences can introduce more useful information to better accomplish the recommendation tasks.

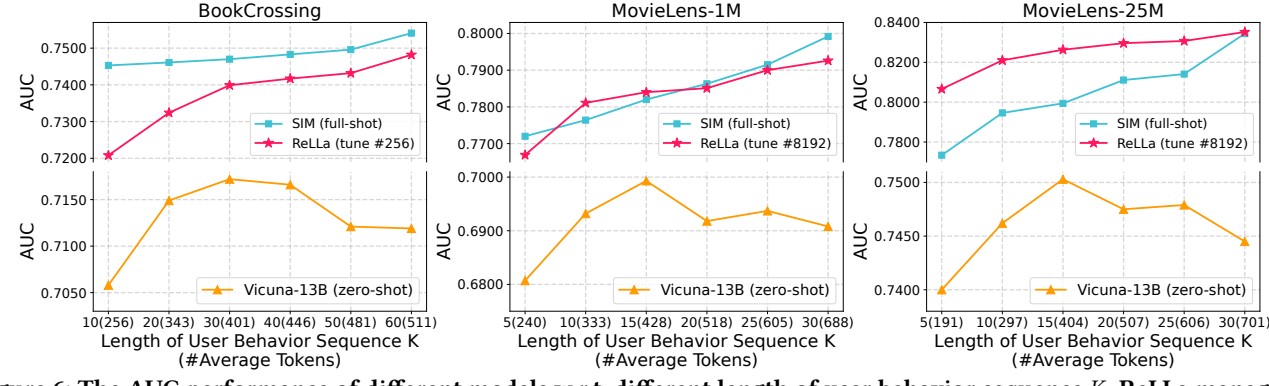

**Figure 6: The AUC performance of different models w.r.t. different length of user behavior sequence $K$. ReLLa manages to mitigate the incomprehension problem of LLMs on recommendation tasks with long user behavior sequences.**

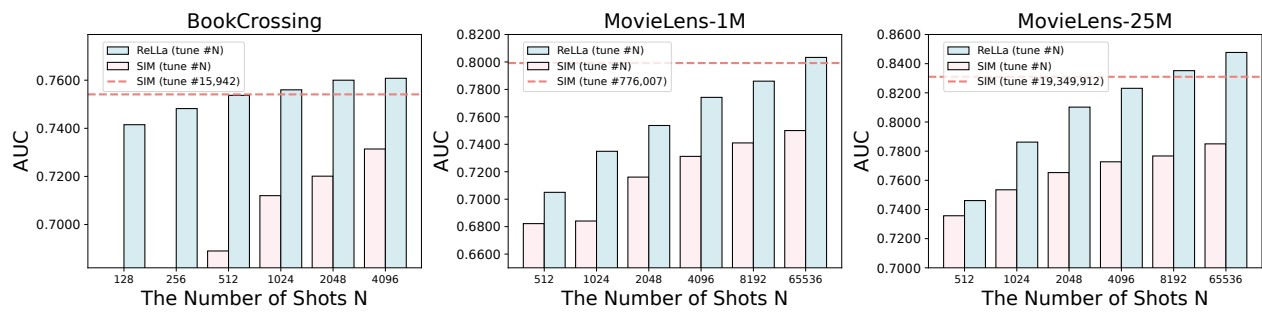

**Figure 7: The AUC performance of ReLLa and SIM w.r.t. different numbers of shots $N$ on three datasets, where "tune #$N$" indicates that we train the model with $N$ training samples. The dashed line denotes the AUC performance of SIM (full-shot) that is trained with the whole training set. Notably, for $N = 128$ and $N = 256$ on BookCrossing dataset, few-shot SIM fails to accomplish the CTR prediction task, where the AUC is merely around 0.5, and is therefore omitted in the figure.**

- However, the performance of Vicuna-13B (zero-shot) only arrives at the peak with $K = 30/15/15$ on BookCrossing/MovieLens-1M/MovieLens-25M datasets, and then starts to decrease with further longer sequence. It is worth noting that the number of involved tokens (*i.e.*, around 500/700/700 for three datasets respectively) is actually far from reaching the context limitation of Vicuna-13B (*i.e.*, 2048 tokens). This indicates that it is non-trivial for LLMs to comprehend the textual context of long behavior sequences for recommendation, where a certain amount of in-domain knowledge is required.
- ReLLa manages to mitigate the incomprehension problem of LLMs on long user behavior sequences for recommendation tasks. Compared with Vicuna-13B (zero-shot), whose performance drops when $K > 30$ on BookCrossing and $K > 15$ on MovieLens-1M and MovieLens-25M, there are no performance turning points for ReLLa. Similar to SIM, the AUC performance of ReLLa achieves continuous improvement as $K$ grows, validating the comprehension ability of ReLLa for the textual contexts with longer behavior sequences.

## 4.4 Data Efficiency (RQ3)

Focusing on few-shot settings, we investigate the data efficiency property by varying the number of shots $N$. In Figure 7, we report the AUC performance of ReLLa and SIM (the best full-shot baseline) with different $N$s. For BookCrossing dataset, $N$ ranges in $\{128, 256, 512, 1024, 2048, 4096\}$. For MovieLens-1M and MovieLens-25M datasets, $N$ ranges in $\{512, 1024, 2048, 4096, 8192, 65536\}$. The

length of user behavior sequence $K$ is set to $K = 60/30/30$ for BookCrossing/MovieLens-1M/MovieLens-25M datasets, respectively.

As depicted in Figure 7, both ReLLa and SIM attain performance enhancement as the number of shots $N$ gradually grows. However, with the same number of shots $N$, ReLLa can outperform SIM significantly and consistently by a large margin. Moreover, when $N$ is extremely small (*e.g.*, 128 and 256) on BookCrossing dataset, SIM even fails to accomplish the CTR prediction task where AUC is merely around 0.5. With limited number of training samples, ReLLa shows remarkable data efficiency property and display considerable few-shot inference ability due to the intrinsic logical reasoning abilities and possession of open-world knowledge of LLMs.

## 4.5 Ablation Study (RQ4)

To analyze the efficacy of each component in our proposed ReLLa framework, we design the following model variants of ReLLa. We set $N = 256/8192/8192$ (<1% setting) and $K = 60/30/30$ for BookCrossing/MovieLens-1M/MovieLens-25M datasets, respectively.

- **ReLLa (Ours)** is the complete version of our proposed method. The training data consists of both original and retrieval-enhanced samples, resulting in a mixed training dataset of $2N$ samples. The testing set only contains pure retrieval-enhanced samples.
- **ReLLa (w/o Mixture)**. We only maintain the retrieval-enhanced data instances to construct the training dataset of $N$ samples. The testing data is still all retrieval-enhanced samples.

**Table 3: The performance of different variants of ReLLa. We remove different components of ReLLa to evaluate the contribution of each part to the model. The best result is given in bold, and the second-best value is underlined.**

| Model Variant | BookCrossing | | | MovieLens-1M | | | MovieLens-25M | | |
|---|---|---|---|---|---|---|---|---|---|
| | AUC | Log Loss | ACC | AUC | Log Loss | ACC | AUC | Log Loss | ACC |
| ReLLa (Ours) | **0.7482** | 0.6265 | **0.6800** | **0.7927** | **0.5475** | **0.7196** | **0.8352** | **0.4693** | **0.7779** |
| ReLLa (w/o Mixture) | 0.7399 | **0.6002** | 0.6715 | 0.7849 | 0.5693 | 0.6985 | 0.8192 | 0.4904 | 0.7715 |
| ReLLa (w/o Retrieval) | 0.7167 | 0.9293 | 0.4898 | 0.7718 | 0.5795 | 0.7039 | 0.8174 | 0.4892 | 0.7685 |
| ReLLa ($\frac{1}{2}N$-shot) | 0.7415 | 0.6268 | 0.6462 | 0.7862 | 0.5781 | 0.6964 | 0.8231 | 0.5157 | 0.7672 |
| ReLLa (w/o IT) | 0.7253 | 0.9277 | 0.5750 | 0.7013 | 0.6250 | 0.6507 | 0.7324 | 0.5858 | 0.7027 |
| ReLLa (w/o IT & Retrieval) | 0.7176 | 0.9507 | 0.5649 | 0.6993 | 0.6291 | 0.6493 | 0.7503 | 0.6308 | 0.6427 |

- **ReLLa (w/o Retrieval)**. We remove the semantic user behavior retrieval for both training and testing samples. That is, training and testing data are all original samples without retrieval enhancements. The training set contains $N$ training samples. This variant indicates the vanilla instruction tuning version over Vicuna-13B, which is similar to TALLRec [1].
- **ReLLa ($\frac{1}{2}N$-shot)**. We halve the number of shots $N$ to $\frac{1}{2}N$, *i.e.*, from 256 to 128 on BookCrossing and from 8192 to 4096 on MovieLens-1M and MovieLens-25M. Therefore, the constructed mixed training set contains $N$ training samples. This variant is intended to decouple and investigate the factors of doubled training samples and pattern enrichment.
- **ReLLa (w/o IT)**. We remove the instruction tuning, while preserving the retrieval-enhanced samples for testing data. This variant indicates the zero-shot version of our proposed ReLLa.
- **ReLLa (w/o IT & Retrieval)**. We remove both the instruction tuning and retrieval operation. Therefore, the testing data only contains original data samples. This variant indicates the zero-shot version of vanilla Vicuna-13B.

The performance of these variants are presented in Table 3, from which we can draw the following observations:

- For ReLLa (w/o Mixture) and ReLLa (w/o Retrieval), their training and testing data comprise exactly the same type of samples, *i.e.*, either pure original samples or retrieval-enhanced samples respectively, which indicates that there is no data inconsistency between the training and testing phases. Nevertheless, both of them significantly underperform our proposed ReLLa by at least 1.12%, 0.99% and 1.95% on BookCrossing, MovieLens-1M and MovieLens-25M in AUC respectively. This highlights the importance of the data mixture strategy, the benefits of which can be broken down into two prominent factors: doubled training samples and pattern enrichment. Doubled training samples lead to a more thorough training process, while pattern enrichment can prevent the model from overfitting and therefore increase the model robustness.
- We introduce the variant ReLLa ($\frac{1}{2}N$-shot) to further decouple and analyze the two factors mentioned above, *i.e.*, doubled training samples and pattern enrichment. Its total number of training samples is the same as those of ReLLa (w/o Mixture) and ReLLa (w/o Retrieval), except that ReLLa ($\frac{1}{2}N$-shot) loses the sight of half truly training instances. In this case, ReLLa ($\frac{1}{2}N$-shot) still outperforms ReLLa (w/o Mixture) and ReLLa (w/o Retrieval) with 0.21%, 0.16% and 0.48% relative AUC improvement, and achieves comparable or better performance in Log Loss and ACC. This

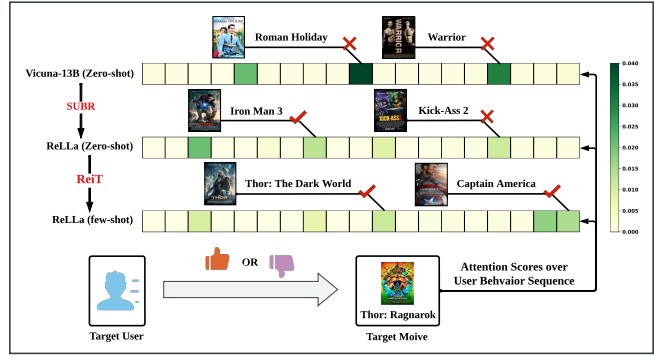

**Figure 8: The case study of ReLLa on MovieLens-25M dataset. We visualize the attention scores over the historical items (*i.e.*, the rectangles) in user behavior sequence at the last hidden layer of LLM. The deeper green a rectangle possesses, the large attention score the corresponding historical item attains, thus contributing more to the final CTR estimation.**

indicates that pattern enrichment as regularization plays a more vital role that contributes to the performance improvement.
- Finally, comparing ReLLa (w/o IT) and ReLLa (w/o IT & Retrieval), which fall back into zero-shot settings, we can observe that ReLLa (w/o IT) generally achieves significant improvements over ReLLa (w/o IT & Retrieval), except for the AUC metric on MovieLens-25M. This demonstrates that semantic user behavior retrieval (SUBR) improves the quality of data samples and makes the filtered behavior sequence more friendly for LLM to extract useful knowledge.

### 4.6 Case Study (RQ5)

In this section, we conduct case study to further analyze how can ReLLa help LLM better understand the long user behavior sequence. As shown in Figure 8, we select a testing sample from MovieLens-25M dataset, and visualize the attention scores of target item over the user behavior sequence at the last hidden layer of three different models (*i.e.*, Vicuna-13B, ReLLa (zero-shot), and ReLLa (few-shot)). The attention score for each historical item is computed by summing up the attention scores of every word token for the textual input of the corresponding item. In Figure 8, each historical item is represented as a rectangle with color ranging from yellow to green. The deeper green a rectangle possesses, the large attention score the corresponding historical item attains, thus contributing more to the final CTR estimation.

For Vicuna-13B (zero-shot), the largest attentions fall on the movie *Roman Holiday* and *Warrior*, which have little relationship with the target movie *Thor: Ragnarok*, and thus the model fails to correctly infer the user's preference towards the target item. Equipped with semantic user behavior retrieval (SUBR), we can reduce the noise of user behavior sequence and bring in more relevant items. As shown in Figure 8, ReLLa (zero-shot) is able to put more attentions to superhero movies (*e.g.*, *Iron Man 3*) that are semantically similar to the target item. However there are still outliers for ReLLa (zero-shot), *e.g.*, the movie *Kick-Ass 2* is generally non-correlated to *Thor: Ragnarok* produced the Marvel. Next, by further applying retrieval-enhanced instruction tuning (ReiT), we can observe that the large attention weights of ReLLa (few-shot) all fall onto relevant superhero movies that are also produced by the Marvel. Therefore, we can conclude that our proposed SUBR and ReiT can help LLM to correctly grasp the correlation between the target item and historical items, thus better comprehending the user behavior sequence.

## 5 RELATED WORK

### 5.1 Traditional CTR Prediction

CTR prediction serves as the key component for various online applications (*e.g.*, recommender systems [78], advertising [56], and web search [15, 19, 45]). It aims to accurately estimate the user's click probability towards a certain target item in a given context [87]. Traditional CTR prediction models can be mainly classified into two categories: (1) feature interaction based models, and (2) sequential recommendation models.

The feature interaction based models generally derive from POLY2 [6] and FM [68]. Their core idea is to capture the second- or high-order feature interaction patterns across multiple feature fields with different operators (*e.g.*, product [7, 23, 63, 77], convolution [47, 81], and attention [70, 80]). For examples, DCN [76], xDeepFM [43], and DCNv2 [77] apply product-based feature crossing operation at each layer for explicit high-order feature interaction modeling. AutoInt [70] and InterHAt [42] adopt the attention mechanism for feature interactions, which provides additional explainable prediction via attention weights.

The sequential recommendation model [58, 91, 92] focuses on user behavior modeling and seeks to dynamically capture users' interests towards a target item according to the given behavior history. They leverage different architectures (*e.g.*, RNN [25, 26], CNN [72], attention [91, 92], memory bank [58, 67]) to handle the user behavior sequence for user preference modeling. For instances, GRU4Rec [26] adopt the gated recurrent unit (GRU) [12] to encode the user's sequential behaviors. Caser [72] introduces the convolution neural network (CNN) to model the union-level patterns among user behavior sequences.

### 5.2 Language Models for Recommendation

As suggested in previous work [44], the adaption of language models to the field of recommender systems can be generally categorized according to the roles they serve in the recommendation pipeline, *i.e.*, feature engineering [3, 5, 11, 37, 50, 55], feature encoder [18, 24, 27, 28, 39, 54, 54, 62, 74, 82, 83, 88], scoring/ranking function [1, 8, 22, 29, 31, 32, 34, 38, 41, 48, 52, 57, 75, 85, 89, 90].

For feature engineering, large language models (LLMs) accept the raw data (*e.g.*, user profiles and item descriptions) as input, and generate supplementary text-based attributes as data augmentations with delicately designed prompts and templates. For example, KAR [79] utilizes the reasoning knowledge on user preferences and the factual knowledge on items by requesting LLMs with factorization prompting techniques. The obtained knowledge can serve as augmented features and promote the recommendation performance in a model-agnostic manner. GENRE [50] employs LLMs to obtain news summarization, synthetic news pieces, and user profiles.

For feature encoder, LLMs are adopted as auxiliary textual feature encoders to (1) enrich the user/item representations with semantic information, and (2) enable cross-domain recommendation with the natural language interface. For instance, U-BERT [62] enhances the user representation by encoding review texts into dense vectors via BERT. UniSRec [28] and VQ-Rec [27] apply a fixed BERT as the encoder for item descriptive texts, in order to achieve unified cross-domain sequential recommendation.

For scoring/ranking function, researchers explore the potential of LLMs to directly serve as the core scoring or ranking module for recommendation, instead of an assistant role for conventional recommendation models (*e.g.*, feature engineering or feature encoder). In this case, LLMs are employed to accomplish either the item scoring task [1, 34, 38, 41, 48, 52, 89, 90], or item generation task [8, 22, 29, 31, 32, 57, 75, 85]. Also, various works [13, 14, 21, 49, 71, 86] attempt to utilize the multi-task capacity of LLMs, and instruct LLMs to solve the multiple tasks (*e.g.*, both scoring and generation) through a unified language interface.

In this paper, we mainly focus on the utilization of LLMs as the scoring/ranking functions, where the pointwise scoring task is adopted for CTR prediction. To the best of our knowledge, we are the first to identify and well formulate the incomprehension problem of LLMs on lifelong user behavior sequences when adopting LLMs for scoring and ranking tasks. A novel ReLLa framework is proposed to mitigate such an issue by introducing the retrieval techniques to promote comprehension ability of LLMs and thus enhance their recommendation performance.

## 6 CONCLUSION

In this paper, we focus on adapting and empowering LLMs as the scoring/ranking function for recommendation tasks. We first identify and formulate the incomprehension problem of LLMs on lifelong sequential behaviors, *i.e.*, LLMs fail to extract useful information from a textual context of long user behavior sequence, even if the length of context is far from reaching the context limitation of LLMs. Hence, we propose a novel ReLLa framework, where semantic user behavior retrieval (SUBR) and retrieval-enhanced instruction tuning (ReiT) are designed to address such an issue and therefore promote the recommendation performance. Extensive experiments validate the effectiveness of our proposed ReLLa compared with existing baselines. Specifically, leveraging only less than 10% training samples, *few-shot* ReLLa can outperform all the *full-shot* traditional CTR models that are trained on the entire training set. This demonstrate the superior data efficiency of ReLLa, as well as its comprehension ability towards long user behavior sequences.

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

## A  DATA PREPROCESSING

Our experiments are conducted on three real-world public datasets (*i.e.*, BookCrossing, MovieLens-1M and MovieLens-25M), and the statistics of the processed datasets are show in Table 1. MovieLens-1M and MovieLens-25M datasets are split into training and testing sets with ratio of 8:1 according to the global timestamp [60]. Since BookCrossing dataset has no timestamps, following previous work [1], we divide it into training and testing sets with ratio of 9:1

by random split of users. Data samples with user behavior sequence length less than 5 are filtered on all three datasets. We describe more preprocessing details as follows:

- **BookCrossing** possesses user-book integer ratings ranging from 0 to 10. We consider samples with rating above 5 as positive, and the rest as negative.
- **MovieLens-1M** contains user-movie integer ratings ranging from 0 to 5. Samples with ratings of 4 and 5 are labeled as positive and the rest as negative. [79, 92]
- **MovieLens-25M** has a scoring range from 0 to 5, with increments of 0.5. We label samples with ratings above 3.0 as positive, and the rest as negative.

Under the few-shot setting with a particular number of shot $N$, we uniformly sample $N$ data instances from the training set, which is then fixed ReLLa during few-shot tuning. Note that the sampled data instances with a smaller $N$ are all included in the sampled few-shot training sets with a larger $N$.

## B BASELINE IMPLEMENTATION

In this section, we describe the hyperparameter configuration for the baseline models from two different categories: (1) traditional CTR models, and (2) LM-based models.

*B.0.1 Traditional CTR Models.* We choose the embedding size from {8, 16, 32} on BookCrossing dataset and {32, 64} on MovieLens-1M and MovieLens-25M datasets. The dropout rate is selected from {0.0, 0.1, 0.2}. The activation function is fixed to ReLU. The learning rate is set to $1 \times 10^{-3}$ and AdamW [51] optimizer is used. On BookCrossing, the batch size is selected from {32, 64}. On MovieLens-1M and MovieLens-25M, the batch size is selected from {256, 512}. More model-specific hyperparameter settings are shown as follows:

- **DeepFM** [23]. On BookCrossing, the size of DNN layer is selected from {32, 64, 128}. The number of DNN layers is selected from {1, 2, 3}. On MovieLens-1M and MovieLens-25M, we choose the size of DNN layer from {128, 256} and the number of DNN layers from {3, 6, 9, 12}.
- **AutoInt** [70]. On BookCrossing, the number of attention layers is selected from {1, 2} and the attention size is set to 32. On MovieLens-1M and MovieLens-25M, the attention layers is selected from {3, 6, 9, 12} and the attention size is selected from {64, 128, 256}. The number of attention heads are all set to 1.
- **DCNv2** [77]. On BookCrossing, the size of DNN layer is selected from {32, 64, 128}. The number of DNN layers and cross layers are selected from {1, 2, 3}. On MovieLens-1M and MovieLens-25M, we choose the size of DNN layers from {128, 256} and the number of DNN layers and cross layers are from {3, 6, 9, 12}.
- **GRU4Rec** [26]. The number of GRU layers is selected from {1, 2, 3}. On BookCrossing, the GRU hidden size and DNN hidden size is selected from {32, 64}. On MovieLens-1M and MovieLens-25M, the GRU hidden size and DNN hidden size is selected from {64, 128, 256}.
- **Caser** [72]. The number of vertical convolution kernels is selected from {2, 4, 8}. The number of horizontal convolution kernels is selected from {4, 8, 16}. The number of DNN layers is selected from {1,2,3}. The DNN hidden size is selected from {32,

64} on BookCrossing and {64, 128, 256} on MovieLens-1M and MovieLens-25M.
- **SASRec** [33]. The number of attention heads is selected from {1, 2, 4}. The number of attention layers is selected from {1, 2, 3}. The attention size is selected from {32, 64, 128} on BookCrossing and {64, 128, 256}. The number of DNN layers is selected from {1,2,3}. The DNN hidden size is selected from {32, 64} on BookCrossing and {64, 128, 256} on MovieLens-1M and MovieLens-25M.
- **DIN** [92]. The number of DIN attention layers and DNN layers are selected from {1, 2, 3}. The DNN hidden size is selected from {32, 64} on BookCrossing and {64, 128, 256} on MovieLens-1M and MovieLens-25M.
- **SIM** [59]. The number of attention layers and DNN layers are selected from {1, 2, 3}. The DNN hidden size is selected from {32, 64} on BookCrossing and {64, 128, 256} on MovieLens-1M and MovieLens-25M.

*B.0.2 LM-based Models.* The structure of the pretrained language models is kept unchanged. And AdamW [51] optimizer is used for all the baselines. The detailed training settings are as follows:

- **CTR-BERT** [53]. We maintain a two-tower model structure based on the BERT [16] model to encode the user and item information respectively. The total number of tuning epochs is set to 10. The batch size is set to 1024. The learning rate is set to $5 \times 10^{-5}$ with linear decay. The warmup ratio is 0.05.
- **P5** [20] is a unified sequence-to-sequence framework with T5 [64] as the backbone pretrained language model for multiple recommendation tasks. In this paper, we leverage P5 for a single task only (*i.e.*, CTR prediction). The total number of epochs is set to 10 with batch size of 32. The learning rate is selected from {$5 \times 10^{-4}, 1 \times 10^{-3}$} with linear decay. The warmup ratio is 0.05. Following P5's official implementation, we also perform gradient clip with threshold equal to 1.0.
- **PTab** [48] adopts the common pretrain-finetune scheme based on the BERT [16] model. PTab first further pretrains the BERT model with the classical masked language modeling objective based on the textualized CTR data, and then finetunes BERT for downstream CTR prediction as a text classification problem. Following the original paper, we pretrain BERT for 10 epochs with batch size equal to 1024. The learning rate for pretraining is set to $5 \times 10^{-5}$ with linear decay. The warmup ratio is 0.05. As for finetuning, the total number of tuning epoch is set to 10 with batch size of 1024. The learning rate for finetuning is initialized at $5 \times 10^{-5}$ with linear decay. The warmup ratio is 0.01.

## C PROMPT ILLUSTRATION

We demonstrate several examples to illustrate the hard prompt templates used for ReLLa on all three datasets.

Figure 9 shows the textual input-output pairs without semantic user behavior retrieval (SUBR), where the user behavior sequence is truncated to most recent $K$ (*e.g.*, $K = 4$ in the figure). As is shown in Figure 10, after applying SUBR, the user behavior sequence will be replaced by most relevant $K$ historical items towards the target item. For example, for MovieLens-25M dataset, historical behaviors retrieved by SUBR are all related to superheros or Marvel, which is highly correlated to the target movie "Thor: Ragnaro". Note that the

---

**BookCrossing**

**Input:**

The user's location is USA. The user's age is 30-34.

The user read the following books in the past and rated them:

['0. Pride and Prejudice (8 stars)', '1. Gone with the Wind (9 stars)', '2. Lolita (5 stars)', '3. Hamlet (7 stars)', '4. Wuthering Heights (8 stars)']

Based on the books the user has read, deduce if the user will like the book ***Jane Eyre***.

Note that more stars the user rated the book, the user liked the book more.

You should ONLY tell me yes or no.

**Output:**

Yes.

---

**MovieLens-1M**

**Input:**

The user is a female. Her job is sales/marketing. Her age is 35-44.

She watched the following movies in order in the past, and rated them:

['0. The Terminator (1984) (3 stars)', '1. Star Wars: Episode IV (1977) (2 stars)', '2. 2001: A Space Odyssey (1968) (3 stars)', '3. Back to the Future (1985) (2 stars)']

Based on the movies she has watched, deduce if she will like the movie ***Aliens (1986)***.

Note that more stars the user rated the movie, the more the user liked the movie.

You should ONLY tell me yes or no.

**Output:**

No.

---

**MovieLens-25M**

**Input:**

The user watched the following movies in the past, and rated them:

['0. Iron Man 3 (2013) (4.0 stars)', '1. Avengers: Age of Ultron (2015) (4.5 stars)', '2. Thor (2011) (4.0 stars)', '3. Thor: The Dark World (2013) (4.0 stars)']

Based on the movies the user has watched, deduce if the user will like the movie ***Thor: Ragnarok (2017)***.

Note that more stars the user rated the movie, the user liked the movie more.

You should ONLY tell me yes or no.

**Output:**

Yes.

---

**Figure 10: Examples of hard prompt templates for three datasets *with* SUBR. The user behavior sequence is constructed by the most relevant $K$ items.**

---

**BookCrossing**

Here is a book. Its title is Jane Eyre. ISBN of the book is 0451523126. The author of the book is Charlotte. The publication year of the book is 1988. Its publisher is Signet Classics.

---

**MovieLens-1M**

Here is a movie. Its title is Aliens (1986). The movie's genre is Fiction.

---

**MovieLens-25M**

Here is a movie. Its title is Thor: Ragnarok (2017). The movie's genre is Action.

---

**Figure 11: Examples of hard prompt templates of item descriptions for three datasets. The textual description is used to obtain the semantic item embedding from LLM, which will then be leveraged by SUBR.**

---

**BookCrossing**

**Input:**

The user's location is USA. The user's age is 30-34.

The user read the following books in the past and rated them:

['0. Hamlet (7 stars)', '1. The King of Torts (6 stars)', '2. Red Storm Rising (5 stars)', '3. The Power of Self-Esteem(2 stars)', '4. Wuthering Heights (8 stars)']

Based on the books the user has read, deduce if the user will like the book ***Jane Eyre***.

Note that more stars the user rated the book, the user liked the book more.

You should ONLY tell me yes or no.

**Output:**

Yes.

---

**MovieLens-1M**

**Input:**

The user is a female. Her job is sales/marketing. Her age is 35-44.

She watched the following movies in order in the past, and rated them:

['0. Gone with the Wind (1939) (5 stars)', '1. The Terminator (1984) (2 stars)', '2. Before Sunrise (1995) (4 stars)', '3. Blade Runner (1982) (3 stars)']

Based on the movies she has watched, deduce if she will like the movie ***Aliens (1986)***.

Note that more stars the user rated the movie, the more the user liked the movie.

You should ONLY tell me yes or no.

**Output:**

No.

---

**MovieLens-25M**

**Input:**

The user watched the following movies in the past, and rated them:

['0. Blade Runner 2049 (2017) (4.5 stars)', '1. Warrior (2011) (3.5 stars)', '2. Brand New Yolki (2017) (1.5 stars)', '3. What We Do in the Shadows (2014) (4.0 stars)']

Based on the movies the user has watched, deduce if the user will like the movie ***Thor: Ragnarok (2017)***.

Note that more stars the user rated the movie, the user liked the movie more.

You should ONLY tell me yes or no.

**Output:**

Yes.

---

**Figure 9: Examples of hard prompt templates for three datasets *without* SUBR. The user behavior sequence is constructed by the most recent $K$ items.**

user behavior sequence generated by SUBR keeps the chronological order in the original lifelong user sequence.

Figure 4 demonstrates how we design prompts for item descriptions on the three datasets, which will be encoded by LLM for semantic user behavior retrieval (SUBR).

