# OpenReview forum: "ReLLa: Retrieval-enhanced Large Language Models for Lifelong Sequential Behavior Comprehension in Recommendation"
_ACM.org/TheWebConf/2024/Conference — TheWebConf24 Oral_

### Official Review · Reviewer_mqgA · 2023-11-14

**Novelty:** 4
**Technical Quality:** 5

**Review:**

The authors claimed that they are the first to identify and well formulate the lifelong sequential behavior incomprehension problem for LLMs in recommendation, where LLMs are generally incomprehensible to a textual context of long user behavior sequence, even if the length of context is far from reaching the context limitation.

The main contribution of the paper is the introduction of a framework called ReLLa. It is designed to mitigate the incomprehension problem of LLMs on long user behavior sequences. The core ideas of this framework include semantic user behavior retrieval (SUBR) to improve the data quality of data samples for zero-shot recommendation and retrieval-enhanced instruction tuning (ReiT) to promote the few- shot recommendation performance with a mixture of original and retrieval-enhanced training samples.

Extensive experiments on three real-world public datasets validate the effectiveness of ReLLa compared with some baselines. It is impressive that this method performs better than full-shot baselines with few-shot settings.

Reasons To Accept:

1. The writing of this paper is clear and easy to understand.
2. The method is simple but effective, achieving competitive or superior performance, especially compared to previous baselines.
3. Choice of datasets is thorough. Tables present full and accurate results, and figures are also expressive.
4. It seems that ReLLa reaches the top with very limited resources compared with other methods.

Reasons To Reject:

1. This paper may not be quite novel. The only core feature introduced to promote the performance is SUBR, which is a simple method used for data augmentation.
2. Empirical results demonstrate the effects of designed augmentation, retrieval and IT. However, more backbone LLMs need to be evaluated.
3. In few-shot settings, even the number of training shots N is selected as <10%, it is still much larger than N values of few-shot settings in other tasks. And ReiT actually feeds twice amount of original data into the model for learning. It is unclear whether the benefits are brought by the LLM itself or proposed methods. Therefore, this setting maybe unfair and should be evaluated and discussed based on more experiments.

Typo:
Line 304: The parentheses are not closed.
Line 929: The content exceeds the space of paper.

After Rebuttal:
I have read the author responses and would like to keep my evaluation scores.

**Questions:**

1. Why the cosine distance can be used to describe the semantic relevance between user behaviors? Have you compared it with other metrics? Please give more theoretical and empirical evidence.

2. Please provide more experimental results on other popular LLMs and strict few-shot settings.

3. Are the SUBR and ReiT specific to recommendation tasks? Can they adapt to tasks of other continual learning tasks like continual NER and continual relation extraction?

**Ethics Review Description:**

There is no ethics issue.

**Reviewer Confidence:**

3: The reviewer is confident but not certain that the evaluation is correct

**Scope:**

4: The work is relevant to the Web and to the track, and is of broad interest to the community

---

### Official Review · Reviewer_jiKX · 2023-11-20

**Novelty:** 5
**Technical Quality:** 6

**Review:**

This paper introduces a novel framework for enhancing the performance of recommender systems using Large Language Models (LLMs).

**Strength**
1. This paper identifies a unique problem in recommendation domains: LLMs' failure to extract useful information from long sequences of user behavior, even when the length of the context is within the model's capability.
2. Instruction fine-tuning has been applied to LLMs for recommendation tasks[1]. The novelty of this work compared to existing works has been justified.
3. The ReLLa exhibits superior performance compared to existing baselines, using fewer data to fine-tune the model.

[1] Bao, Keqin, et al. "Tallrec: An effective and efficient tuning framework to align large language model with recommendation." RecSys '23

**Weakness**
1. When applied in an online recommendation system, will the model meet the efficient request? Does the authors propose any solution to speed up the inference?

**Questions:**

See weakness

**Ethics Review Description:**

No issue

**Reviewer Confidence:**

3: The reviewer is confident but not certain that the evaluation is correct

**Scope:**

4: The work is relevant to the Web and to the track, and is of broad interest to the community

---

### Official Review · Reviewer_uhND · 2023-11-23

**Novelty:** 3
**Technical Quality:** 3

**Review:**

### Summary
For recommendation, since each user's historical data is often lengthy, existing LLM-powered methods may not utilize the full historical data, and extracting useful information from it with LLMs is difficult. Inspired by this motivation, this work aims to retrieve only the relevant historical data and then enhance LLMs only with it for recommendation. Also, the authors propose to annotate the recommendation data, which consists of the recent items (input) and the target object to predict (output), by replacing the recent items with relevant items from the user's historical data obtained by retrieval. The authors show that the proposed ReLLa outperforms existing methods with two different scenarios: without training (zero-shot) or with the training of a few samples (few-shot).

---

### Strengths
* The motivation to use the retrieval to select and inject only the relevant items from the historical data, instead of injecting only the recent items, is clear and well investigated.
* The idea of performing the data augmentation over the few training samples with retrieval is interesting and reasonable.
* This paper is very well-written and easy to follow.

---

### Weaknesses
* The main proposed idea of retrieving only the relevant items from the historical data for recommendation is generally utilized in recent literature [A, B, C].
* The usage of LLMs for the click-through rate (CTR) prediction task is not convincing enough. In particular, in order to predict the likelihood of clicking one item, the proposed approach requires one forward pass of the LLM. In other words, for k samples, it requires k forward passes of the LLM, which might be very slow in production, compared to the existing traditional and smaller LM-based approaches.
* The comparisons between different methods in the main table (Table 2) are not fair enough. In particular, for the baseline methods, the authors use the smaller LMs (Bert or T5), while, for the proposed method, the authors use the multi-billion LM (13B). In this regard, the most performance gains of the proposed method compared to the existing methods might come from using the powerful LLM, not from using the proposed retrieval and data augmentation strategies.
* The authors validate the proposed method over the few-shot setting, i.e., comparing it with the baseline methods trained with the full dataset. In this regard, the authors may also fine-tune the proposed method with the full dataset and then make the apple-to-apple comparisons between different methods. And, the justification for this experimental setup (full fine-tuning for baselines vs few-shot for the proposed method) is not clear.

---

[A] LaMP: When Large Language Models Meet Personalization, 2023.

[B] RecMind: Large Language Model Powered Agent For Recommendation, 2023.

[C] Memory-Augmented LLM Personalization with Short- and Long-Term Memory Coordination, 2023.

**Questions:**

Please see the main review above. The points below are minor suggestions/questions.
* In Section 3.3, the authors repeatedly argue that the proposed data augmentation strategy alleviates the overfitting issue of LLM training, which further has the advantage of generalization. However, these two claims are not supported by experimental evidence. I would like to encourage authors to include explanations and perhaps some experimental analyses for it.
* In Table 2, the authors report the results with p-value. In this regard, it might be also valuable to provide the standard deviations of each result. Also, I am wondering how many runs are performed to conduct the t-test and to derive the p-value.

**Reviewer Confidence:**

4: The reviewer is certain that the evaluation is correct and very familiar with the relevant literature

**Scope:**

3: The work is somewhat relevant to the Web and to the track, and is of narrow interest to a sub-community

---

### Official Review · Reviewer_r3wK · 2023-11-24

**Novelty:** 6
**Technical Quality:** 6

**Review:**

This paper introduces ReLLa, a framework that enhances large language models (LLMs) for zero-shot and few-shot recommendation tasks. It addresses the lifelong sequential behavior incomprehension problem by using semantic user behavior retrieval (SUBR) for zero-shot recommendation and retrieval-enhanced instruction tuning (ReiT) for few-shot recommendation. ReLLa outperforms existing models on real-world datasets, even with limited training samples, demonstrating its effectiveness in recommendation tasks. The paper is notable for its strong motivation, clear and well-structured presentation, and the availability of source code for easy experiment replication. The results demonstrate that the proposed framework significantly improves click-through rate (CTR) recommendation performance compared to several state-of-the-art baselines, highlighting the modules' effectiveness in enabling LLMs to extract valuable information from lengthy user behavior sequences.

**Questions:**

1. In Section 4.4, could the authors provide a more in-depth analysis of why Vicuna-13B only peaks at 𝐾 = 15 and fails to extract useful information with longer sequences? If possible, would increase the model size, for example, to a 70B model, exhibit the same phenomenon?

2. In the zero-shot recommendation setting, how was the dimensionality for PCA chosen?

3. Could the authors provide a comparison of the model size and time complexity of ReLLa relative to previous SOTA models like SIM?

**Reviewer Confidence:**

4: The reviewer is certain that the evaluation is correct and very familiar with the relevant literature

**Scope:**

4: The work is relevant to the Web and to the track, and is of broad interest to the community

---

### Official Review · Reviewer_Yba1 · 2023-11-28

**Novelty:** 5
**Technical Quality:** 4

**Review:**

The paper introduces ReLLa (Retrieval-enhanced Large Language Models), a framework designed to improve recommender systems using Large Language Models (LLMs) in zero-shot and few-shot recommendation settings. It addresses the lifelong sequential behavior incomprehension problem, where LLMs, exemplified by vicuna, struggle to efficiently process extended user behavior sequences. ReLLa employs two main strategies: Semantic User Behavior Retrieval (SUBR) for zero-shot recommendations and Retrieval-enhanced Instruction Tuning (ReiT) for few-shot scenarios.

Quality and Clarity: The paper is well-organized, offering a distinct problem definition, methodological approach, and empirical validation. The issue of sequential behavior incomprehension in LLMs and ReLLa's solution is clearly and effectively communicated.

Originality: The methodology presented by ReLLa in addressing LLMs' challenges in recommender systems is innovative. SUBR and ReiT are uniquely combined to enhance LLMs' performance in recommendation-related tasks.

Significance: ReLLa's approach is particularly relevant for recommender systems, showcasing efficiency in handling zero-shot and few-shot learning scenarios. Its capability to surpass traditional models using fewer training samples is significant, potentially impacting the field greatly.

Pros

1. Novel method to improve LLMs in recommender systems.

2. Shows enhanced performance over conventional models in few-shot settings.

Cons

1. The discussion on the lifelong sequential behavior incomprehension problem primarily focuses on vicuna, leaving uncertainty about its applicability to other LLMs. Further validation across various LLMs is needed.

2. While ReLLa's superiority in few-shot scenarios over traditional full-shot methods is noted, there is a lack of detailed complexity analysis. The computational costs associated with these improvements remain unclear. Are there any specific computational requirements or limitations for implementing ReLLa in practical recommender systems?

3. The paper would benefit from a more extensive exploration of its limitations and potential biases.

**Questions:**

Please refer to above section.

**Reviewer Confidence:**

4: The reviewer is certain that the evaluation is correct and very familiar with the relevant literature

**Scope:**

4: The work is relevant to the Web and to the track, and is of broad interest to the community

---

### Decision · Program_Chairs · 2024-01-22

**Decision:**

Accept (Oral)

**Comment:**

The paper proposes a novel framework to improve recommender systems using Large Language Models in zero-shot and few-shot recommendation settings by addressing the lifelong sequential behavior incomprehension problem. The idea is novel. The experiments clearly demonstrate the improved performance. The paper is well written and easy to follow.